# CT-Diagnosed Sarcopenia and Cardiovascular Biomarkers in Patients Undergoing Transcatheter Aortic Valve Replacement: Is It Possible to Predict Muscle Loss Based on Laboratory Tests?—A Multicentric Retrospective Analysis

**DOI:** 10.3390/jpm12091453

**Published:** 2022-09-04

**Authors:** Stefan Hecht, Elke Boxhammer, Reinhard Kaufmann, Bernhard Scharinger, Christian Reiter, Jürgen Kammler, Jörg Kellermair, Matthias Hammerer, Hermann Blessberger, Clemens Steinwender, Uta C. Hoppe, Klaus Hergan, Michael Lichtenauer

**Affiliations:** 1Department of Radiology, Paracelsus Medical University of Salzburg, 5020 Salzburg, Austria; 2Department of Internal Medicine II, Division of Cardiology, Paracelsus Medical University of Salzburg, 5020 Salzburg, Austria; 3Department of Cardiology, Medical Faculty of the Johannes Kepler University Linz, 4020 Linz, Austria

**Keywords:** aortic valve stenosis, biomarker, PMAi, sarcopenia, TAVR

## Abstract

Background: Patients with severe aortic valve stenosis (AS) often present with heart failure and sarcopenia. Sarcopenia, described as progressive degradation of skeletal muscle mass, has frequently been implicated as a cause of increased mortality, prolonged hospitalization and generalized poor outcome after transcatheter aortic valve replacement (TAVR). At present, sarcopenia is defined by the European Working Group on Sarcopenia in Older People (EWGSOP) based on clinical examination criteria and radiological imaging. The aim of the present study was to compare patients with Computed Tomography (CT)-diagnosed sarcopenia with regard to the expression of cardiovascular biomarkers in order to obtain additional, laboratory-chemical information. Methods: A total of 179 patients with severe AS were included in this retrospective study. Sarcopenia was determined via CT by measurement of the psoas muscle area (PMA), which was indexed to body surface area (PMAi). According to previous studies, the lowest tertile was defined as sarcopenic. Patients with (59/179) and without sarcopenia (120/179) in the overall cohort were compared by gender-specific cut-offs with regard to the expression of cardiovascular biomarkers such as brain natriuretic peptide (BNP), soluble suppression of tumorigenicity-2 (sST2), growth/differentiation of factor-15 (GDF-15), heart-type fatty-acid binding protein (H-FABP), insulin like growth factor binding protein 2 (IGF-BP2) and soluble urokinase-type plasminogen activator receptor (suPAR). Additionally, binary logistic regression analyses were calculated to detect possible predictors of the presence of sarcopenia. Results: No statistical differences regarding one-year survival could be detected between sarcopenic and non-sarcopenic patients in survival curves (log rank test *p* = 0.179). In the entire cohort, only BNP and hemoglobin (HB) showed a statistically significant difference, with only HB emerging as a relevant predictor for the presence of sarcopenia after binary logistic regression analysis (*p* = 0.015). No relevant difference in biomarker expression could be found in the male cohort. Regarding the female cohort, statistically significant differences were found in BNP, HB and hematocrit (HK). In binary logistic regression, however, none of the investigated criteria could be related to sarcopenia. Conclusion: Regardless of gender, patients with imaging-based muscle degradation did not demonstrate significantly different cardiovascular biomarker expression compared to those without it.

## 1. Introduction

Sarcopenia is described as progressive skeletal muscle wasting. Notwithstanding underlying neurological diseases, in most cases, it occurs in the elderly. Epidemiologically, between 11% and 50% of patients ≥ 80 years of age have relevant sarcopenia [1], which leads to an increased risk of fractures, reduced mobility, decreased quality of life and increased incidence of underlying respiratory and cardiac diseases [2].

The current definition of sarcopenia was established by the European Working Group on Sarcopenia in Older People (EWGSOP) in 2010 [3] and revised in 2018 [4]. In addition to practicable clinical examination tools such as the timed-up-and-go test and the gait speed or chair stand test, relevant radiological criteria obtained by X-ray, computed tomography (CT) and even magnetic resonance imaging are included in this definition. An important lead structure for the radiological definition of sarcopenia is, among others, the psoas muscle. CT images, usually acquired during routine workups (e.g., preoperatively), and not for skeletal muscle loss definition alone, can be used to measure the psoas muscle area and/or diameter [5].

Patients with severe aortic valve stenosis (AS) are commonly older than 80 years of age and are significantly limited in terms of mobility. Although the AS typical triad, consisting of angina pectoris, dyspnea and syncope, is only very rarely experienced in this combination, the occurrence of one of these symptoms is considered symptomatic AS, which significantly limits patient survival [6]. More than 20 years ago, severe AS in patients of advanced age was treated conservatively, since patients could no longer be referred to surgical intervention due to their age and the presence of concomitant diseases. Nowadays, transcatheter aortic valve replacement (TAVR) is an effective, minimally invasive intervention for the replacement of the severely sclerosed and almost nonfunctional aortic valve [7].

Several previous studies reported that sarcopenia in TAVR patients was a relevant predictor of mortality, prolonged length of hospital stay and functional decline [8]. Definitive radiological cut-off values for the presence of sarcopenia—in this case, PMAi—are lacking, as the existing literature defines them individually depending on gender using the lowest tertiles [9], or, more rarely, quartiles [10]. In this study, an attempt was therefore made to assign the presence or absence of sarcopenia to patients on the basis of PMAi and to examine these two groups with regard to the expression of multiple cardiovascular biomarkers in order to be able to make laboratory-chemical observations about sarcopenia.

## 2. Material & Methods

### 2.1. Study Population

Originally, 221 transfemoral TAVR had been performed between 2016 and 2018 at University Hospital Salzburg and University Hospital Linz. Forty-two patients had to be excluded because of missing weight and height information, because CT data of lumbar spine were not available or because the quality of the CT scan was not sufficient. Finally, 179 patients were recommended for inclusion in the study.

Data analyses were performed in accordance with the principles of the Declaration of Helsinki and good clinical practice. Approval of the study protocol was done by the local ethics committees of the Paracelsus Medical University Salzburg (415-E/1969/5-2016) and the Johannes Kepler University Linz (E-41-16). Written informed consent for study participation was obtained from all patients.

### 2.2. Transthoracic Echocardiography

Transthoracic echocardiography was performed using common ultrasound devices (iE33 and Epiq 5; Philips Healthcare, Hamburg, Germany). Severe AS was categorized according to current guidelines of the European Society for Cardiology. AV Vmax (maximal velocity over aortic valve) of 4.0 m/s, AV dpmean (mean pressure gradient over aortic valve) ≥40 mmHg and an aortic valve area ≤1.0 cm^2^ defined severe AS. Left ventricular ejection fraction (LVEF) was measured by Simpson’s method. To graduate mitral, aortic and tricuspid valve regurgitation in minimal, mild (I), moderate (II) and severe (III) cases, spectral and color-Doppler images were used.

### 2.3. CTA Protocol and Measurement of PMAi for Sarcopenia Assessment

At both cardiological centers, patients received a pre-interventional, ECG triggered CT-angiography (CTA) of the entire aorta including the femoral arteries in order to assess, among others, aortic annulus size, aortic, coronary and general vascular anatomy and vascular access side for routine diagnostic workup. Two radiologists, one board certified with seven years of experience in vascular imaging (radiologist 1) and the other in the fourth year of training (radiologist 2), independently assessed the psoas muscle area (PMA: mm^2^). The radiologists were blinded to all clinical and laboratory data. Measurements were obtained on axial CTA scans of the entire aorta performed on a multidetector CT scanner with a patient size-adapted tube voltage (80–120 kVp) and active tube current modulation. A bolus-tracking technique was applied with a 100 mL bolus of non-ionic iodinated contrast media followed by 70 mL saline solution injected at a flow rate of 3.5–5 mL/s. Imaging data were analyzed using a soft tissue kernel with a slice thickness of 3 mm and a reconstruction interval of 2 mm. PMA was measured in all patients at the level of the superior endplate of the third lumbar vertebrae (L3) by manually outlining the right and left psoas muscles (Figure 1 and Figure 2). The total area of the psoas muscle was obtained by adding the areas of the left and right psoas muscles. Normalization to body surface area (BSA) yielded the indexed PMA (PMAi: mm^2^/m^2^). Measurements were performed on a picture archiving and communication system (PACS, Workstation, Impax; Agfa, Mortsel, Belgium).

The psoas muscle was used not only because of its good comparability with other sarcopenia studies, which also used identical radiological parameters, but also with regard to its physiological aspect. The psoas musculature plays a crucial role in locomotion, upright gait as well as stair climbing. Since elderly patients with severe AS are very limited in their range of action due to respiratory distress, their mobility decreases, and with it, the muscle mass of this important muscular apparatus.

### 2.4. Biomarker Analysis

On the day of hospitalization, and thus, one day before the TAVR procedure, blood samples were obtained using a vacuum-containing system. For blood collection, patients were in a fasting state, in a sitting position and without preceding exercise. Collection tubes were centrifuged and plasma was separated from blood components and then frozen at −80 °C. All 179 samples were measured at similar time points.

Plasma levels of soluble suppression of tumorigenicity-2 (sST2: pg/mL), growth/differentiation of factor-15 (GDF-15: pg/mL), heart-type fatty-acid binding protein (H-FABP: ng/mL), insulin like growth factor binding protein 2 (IGF-BP2: pg/mL) and soluble urokinase-type plasminogen activator receptor (suPAR: pg/mL) were measured twice using enzyme-linked immunosorbent assay (ELISA) kits (sST2: Duoset DY523, GDF-15: Duoset DY957, H-FABP: Duoset DY1678, IGF-BP2: Duoset DY674 and suPAR: Duoset DY807; R&D Systems, Minneapolis, MN, USA). The mean value from both measurements was used for the final analysis. Instructions were followed as described by the manufacturer. ELISA plates (Nunc MaxiSorp flat-bottom 96 well plates, VWR International GmbH, Vienna, Austria) were pretreated with a diluted capture antibody overnight. After appropriate washing processes and blocking with a reagent diluent, the plates were prepared for the final assay procedure. After putting serum samples and standard protein onto the wells of ELISA plates (Nunc MaxiSorp flat-bottom 96 well plates, VWR International GmbH, Vienna, Austria), the probes were incubated for two hours. Afterward, plates were washed and treated with Tween 20/PBS solution (Sigma Aldrich, St. Louis, MO, USA). A biotin-labeled antibody (detection antibody) was placed in a reagent diluent and was added for another two hours. A further washing process was performed, and the probes were treated with streptavidin–horseradish peroxidase solution. A color reaction was generated after adding tetramethylbenzidine (TMB; Sigma Aldrich, St. Louis, MO, USA). Optical density was determined at 450 nm on an ELISA plate reader (iMark Microplate Absorbance Reader, Bio-Rad Laboratories, Vienna, Austria).

Many of these biomarkers examined in this study have already been studied in a similar context regarding heart failure and cardiac cachexia and were compiled in a review by Loncar et al. [11].

### 2.5. TAVR Procedure

Indications for TAVR were assessed by a multidisciplinary heart team consisting of cardiologists and cardiologic surgeons. The TAVR procedure was performed as previously described [12]. All 179 patients received TAVR via transfemoral access with TAVR devices of second generation (CoreValve™ Evolut™ R; Medtronic Inc., Minneapolis, MN, USA) or third generation (CoreValve™ Evolut™ Pro; Medtronic Inc., Minneapolis, MN, USA).

### 2.6. Statistical Analysis

Statistical analysis and graphical representations were performed using SPSS (Version 25.0, SPSSS Inc., Armonk, NY, USA).

Linear correlation statistic to calculate the Pearson correlations coefficient (r) was performed to investigate correlations of PMA with factors related to body constitution. PMA (mm^2^) correlated considerably better with BSA (m^2^) according to the Dubois formula (r = 0.568) than with body mass index (BMI: kg/m^2^) (r = 0.217) in our study. Therefore, PMA was divided by BSA, creating indexed PMA (PMAi: mm^2^/m^2^). Since PMAi is gender-dependent, it was divided into tertiles according to male and female sex. Patients in the lowest tertile were referred to as sarcopenic.

A Kolmogorov-Smirnov test was carried out to test variables for normal distribution. Normally distributed metric data were expressed as mean ± standard deviation (SD) and analyzed using an unpaired student’s *t*-test. Not-normally distributed metric data were expressed as median and interquartile range (IQR). The Mann-Whitney-U-test was applied for statistical analyses between sarcopenic and non-sarcopenic patients. Frequencies/percentages were used for categorial data and compared using the chi-square test.

A Kaplan-Meier curve with corresponding log-rank test was generated to determine whether there were differences in one-year survival between patients with radiologically proven sarcopenia compared to those without.

Expressions of biomarker plasma levels were statistically compared based on the two groups, i.e., “sarcopenia” vs. “no sarcopenia”. This was additionally examined on a gender-specific basis. To recognize possible influencing factors regarding the presence of sarcopenia, a univariate binary logistic regression analysis was first completed. For better comparability, a Z transform was absolved for metric data. Subsequently, multivariate binary logistic regression was performed to assess the independent factors regarding the prediction of sarcopenia. Therefore, the covariates associated with the detection of sarcopenia in the univariate analysis (*p* = 0.100) were entered, and a backward variable elimination was carried out.

A *p*-value (two-tailed test) <0.050 was considered statistically significant.

## 3. Results

### 3.1. Study Cohort and Baseline Characteristics

In males, the cut-off value for PMAi between the first and second tertiles, and thus, with respect to sarcopenia, was ≤757.16 mm^2^/m^2^, whereas in females, the cut-off value measured ≤ 569.88 mm^2^/m^2^.

Fifty-nine patients, i.e., nearly 33.0% (lowest tertile of PMAi) fulfilled the radiological criteria of sarcopenia. Among them, 52.5% were men and had a mean age of 83.03 ± 4.68 years. With the exception of height, brain natriuretic peptide (BNP) and hemoglobin (HB), sarcopenic and non-sarcopenic patients presented no significant differences in baseline characteristics (Table 1).

### 3.2. Kaplan-Meier Results

A Kaplan-Meier curve was performed with regard to one-year survival based on the absence or presence of sarcopenia (Figure 3).

Within the sarcopenia group, 15/59 patients died within one year after TAVR, which corresponded to a percentage of 25.4%. In contrast, 42/120 patients from the non-sarcopenia group (35.0%) were no longer alive after one year. The log-rank test performed showed no statistical significance with *p* = 0.179.

### 3.3. Gender-Independent Biomarker Concentrations According to Radiological Absence or Presence of Sarcopenia

Biomarker concentrations were calculated for sarcopenic and non-sarcopenic patients without consideration of gender. The results are presented in Table 1 (section: laboratory data) and in Figure 4.

Among the analyzed biomarkers, only HB and hematocrit (HK) showed lower plasma levels in sarcopenic patients compared to non-sarcopenic patients (HB: 12.40 ± 2.90 g/dL vs. 13.05 ± 2.17 g/dL; HK: 37.20 ± 7.70% vs. 38.80 ± 6.10%). However, a statistically significant difference between both groups could only be calculated for HB (*p* = 0.024).

For the remaining biomarkers, i.e., BNP, sST2, GDF-15, H-FABP, IGF-BP2 and suPAR, plasma levels were marginally elevated in the sarcopenic study group compared to the non-sarcopenic study group. The only significantly different expression was also found on the side of BNP (1906.00 ± 2421.08 pg/mL vs. 838.00 ± 1887.75 pg/mL; *p* = 0.010).

### 3.4. Gender-Independent Binary Logistic Regression Regarding the Prediction of Sarcopenia

To filter out potential influencing factors for the presence of sarcopenia, saturable baseline characteristics were examined in a binary logistic regression analysis. The results are presented in Table 2.

In our univariate, gender-independent analysis, height, BMI, aortic valve insufficiency, tricuspid valve insufficiency, HB, HK and BNP showed a *p* ≤ 0.100 and were therefore included in a multivariate analysis with backward elimination. Here, only HB revealed statistical significance, with a hazard ratio (95% confidence interval) of 0.615 (0.416–0.908) and a *p* = 0.015.

### 3.5. Gender-Dependent Biomarker Concentrations in Dependence of Radiological Absence or Presence of Sarcopenia

A separate analysis of established biomarkers divided into male and female gender was performed in Table 3.

In males, no significant differences could be detected between sarcopenic and non-sarcopenic subjects. However, in females, significantly higher plasma levels of BNP (1562.50 ± 2450.00 pg/mL vs. 758.00 ± 980.70 pg/mL; *p* = 0.007) and significantly lower plasma levels of HB (11.90 ± 1.88 g/dL vs. 12.70 ± 2.20 g/dL; *p* = 0.008) and HK (36.15 ± 5.18% vs. 37.55 ± 5.28%; *p* = 0.023) were found in the sarcopenic group.

### 3.6. Gender-Dependent Binary Logistic Regression Regarding the Prediction of Sarcopenia

The already known binary logistic regression was calculated a second time separately for male (Table 4) and female gender (Table 5).

In males, the only criterion that was found to have a significant influence on the radiologically determined sarcopenia by PMAi was body size. All other baseline characteristics examined were not significantly related.

In the female cohort, the STSScore, HB, HK and BNP had a *p* ≤ 0.100 in the univariate analysis. Finally, in the multivariate calculation, none of the univariate factors revealed a significant value.

## 4. Discussion

In the western world, age structures are shifting toward an older population, and thus, the prevalence of sarcopenia and frailty syndrome is steadily increasing. For the assessment of frailty, there are now proven scores [13] which are already routinely used in clinical settings. Regarding sarcopenia, the EWGSOP has already published numerous clinical tests as well as radiological criteria to make this clinical picture more tangible and understandable [3,4].

Nevertheless, for both frailty syndrome and sarcopenia, there is a lack of adequate laboratory parameters that could underpin the likelihood of their presence and thus optimize risk assessment and prevention.

Therefore, the aim of this study in patients with severe AS was to investigate sarcopenia, which was graded by radiological criteria via measurement of the PMAi, with respect to common cardiovascular biomarkers.

## 5. No Difference in Survival between Sarcopenic and Non-Sarcopenic Patients

In the Kaplan-Meier curve, there was no significant difference in one-year survival between patients with radiological sarcopenia and those without. Additionally, in a separate, not explicitly presented classification into the respective tertiles, no statistically significant difference was detectable between the three groups. This result is in clear contradiction with previous studies by Kofler et al. [14] and Saji et al. [15].

The main reason for these contrasting results is most likely to be found in the inclusion period. Kofler et al. [14] included patients from 2010 to 2017, while Saji et al. [15] included patients from 2009 to 2015. In both cohorts, TAVR valves were implanted, especially of the first and partly also of the second generation, whereas in our cohort, mainly second-generation (CoreValve™ Evolut™ R) and even third-generation TAVR valves (CoreValve™ Evolut™ Pro) were implanted. Further studies have already shown that early generations of TAVR valves had higher complication rates than the currently deployed generation, such as increased embolic events, bleeding events or severe paravalvular regurgitations [16,17]. This could definitely be causal for the different Kaplan-Meier curves in the context of one-year survival.

Also, it should be noted that the cut-off values determined in this work were higher than those obtained for PMAi in the cohort of Kofler et al. [14] (lowest tertile men ≤ 660 mm^2^/m^2^; lowest tertile women ≤ 498 mm^2^/m^2^) because of the lack of definitive cut-off values. Even using the values of Kofler et al. (data not shown) as a cut-off for sarcopenia, a log-rank test of *p* = 0.235 resulted in no statistical significance.

Therefore, in times of considerable technical progress, significantly improved follow-up and steadily increasing experience of interventionalists, the extent to which sarcopenia is actually still life-limiting should be evaluated on a larger study population.

## 6. Hemoglobin—“Old but Gold”?

In the entire gender-specific cohort, only BNP and HB showed statistically significant differences, with BNP increased and HB decreased in the sarcopenic TAVR patients. Other cardiovascular biomarkers examined, i.e., sST2, GDF-15, H-FABP, IGF-BP2 or suPAR, did not demonstrate relevant differences between the two groups studied. Finally, in the context of binary logistic regression, HB remained a relevant criterion for the presence or absence of sarcopenia. These results were congruent with those of a recently published study by Tseng et al. [18], which demonstrated an independent association between HB levels and sarcopenia in 730 participants. For example, patients with low HB levels (cut-off men: <13 g/dL; cut-off women: <12 g/dL) had significantly worse gait speed and weaker handgrip strength test results. Additionally, in an Australian cohort by Bani Hassan et al. [19], a relevant correlation between sarcopenia, according to the EWGOP criteria, and anemia could be established. The cut-off values for the presence of anemia corresponded to the same criteria as those used by Tseng et al. [18]. The cause of anemia in severe AS patients is multifactorial. One cause may be Heyde syndrome, first described in 1958, with the triad of aortic stenosis, gastrointestinal bleeding due to angiodysplasia of the gastrointestinal tract and acquired von Willebrand syndrome [20]. The prevalence of Heyde syndrome is reported in the literature to be only 1–3% [21], but the number of patients with alterations in von-Willebrand factor (reduction in high-molecular-weight multimers) is likely to be 20% to 70% [22], which may further increase the risk of bleeding with (often) additional atrial fibrillation under oral anticoagulation. Another important factor is certainly malnutrition or the malabsorption of basic substances that are crucial for hematopoiesis, such as iron, folic acid and vitamin B12. Using a cohort of 300 patients with AS at an average age of 83.8 ± 0.5 years, Fukui et al. [23] showed that 11.3% of patients had definite malnutrition and 42.3% were on the borderline of malnutrition. As shown here, the causes of anemia are diverse and can be microcytic/hypochromic, normocytic/normochromic, or macrocytic/hyperchromic in origin. Ultimately, any form of anemia results in an undersupply of oxygen to organs and tissues. This hypoxia is further exacerbated when one considers that in the study by Fukui et al. [23], malnutrition in patients with severe AS was closely associated with characteristics such as higher NYHA stage or low left ventricular ejection fraction (LVEF). Since both oxygen and iron are essential substances for the muscle protein myoglobin, it is not surprising that malnutrition and associated anemia lead to progressive skeletal muscle breakdown. This pathophysiology can be very well reconciled with the findings of the working group of Hsu et al. [24], who, in turn, demonstrated that there was a close association between malnutrition and sarcopenia in patients prior to TAVR.

## 7. Cardiovascular Biomarkers in Sarcopenia—Pathophysiology Too Complex for Adequate Conclusion?

In the initial overview of the total cohort, BNP emerged as a potential factor for the presence of sarcopenia before multivariate analysis, which led its exclusion. Other generally common cardiac markers such as sST2 or H-FABP remained without seminal significance for sarcopenia. Considering gender, no analyzed biomarker with statistically significant difference was detectable in the male cohort. In the female cohort, there were statistically significant differences in BNP, HB and HK, but all biomarkers and other baseline characteristics were not significantly related to sarcopenia when binary logistic regression was used. Heart failure based on a valvular cause, such as severe AS in this case, is often accompanied by cardiac cachexia and, as a consequence, sarcopenia [25,26]. Previous studies have failed to identify a relevant biomarker to confirm the diagnosis of sarcopenia. Accordingly, the results in this study are also not encouraging. The pathophysiological processes occurring due to severe AS, starting with hypertrophy of the myocardium and continuing with the progressive reduction of cardiac function, the onset of anemia with tissue hypoxia and the resulting cardiac cachexia or sarcopenia, may be too complex to allow the identification pure sarcopenia on the basis of single biomarkers [11]. Therefore, the clinical picture of the patient, combined with radiological criteria and appropriate functional tests, as proposed by EWGOP, remain the most viable approach.

## 8. Conclusions

Patients with radiologically verified sarcopenia by PMAi showed no significant correlation with respect to the expression of cardiovascular biomarkers compared with patients without radiologically detectable sarcopenia. Therefore, the search for a suitable laboratory biomarker to detect sarcopenia in TAVR patients continues.

## 9. Limitation

The present study is based on data from a small cohort over a circumscribed time period (2016–2018). The small number of patients may have exaggerated the importance of outliers. Biomarker levels were only measured at baseline without observations regarding expression after TAVR procedure. Additionally, technical pitfalls in echocardiographic and radiological measurements which lead to misclassifications should always be conceded, even if all examinations were performed by experienced clinical investigators.

## Figures and Tables

**Figure 1 jpm-12-01453-f001:**
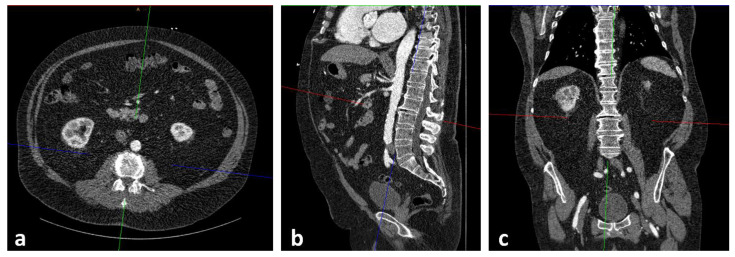
Process of slice selection using the Extended Multiplanar Reconstruction Plugin in IMPAX. The axial slice (**a**) was aligned parallel to the superior endplate of the third lumbar vertebrae at sagittal (**b**) and coronal (**c**) reconstructions.

**Figure 2 jpm-12-01453-f002:**
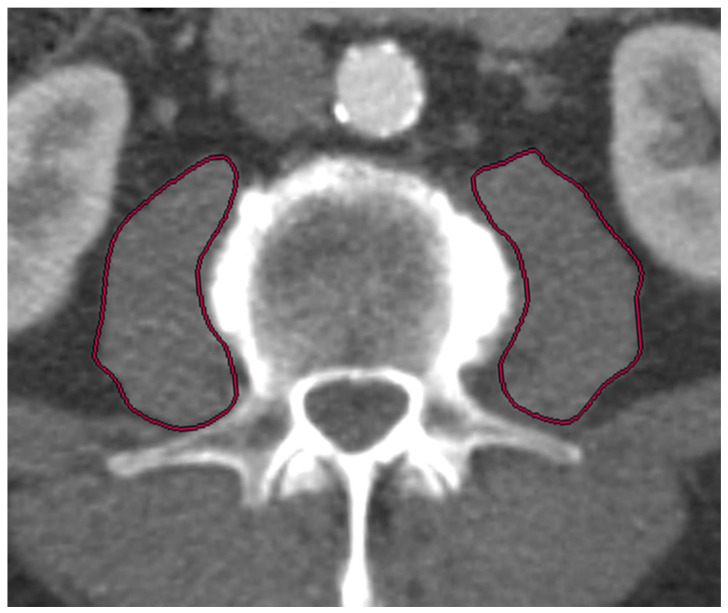
Measurement of left and right Psoas muscle area (PMA) by manually outlining the psoas muscle perimeter (different patient to Figure 1) on an axial slice obtained as demonstrated in Figure 1. The sum of left and right muscle areas normalized to body surface area yielded PMAi.

**Figure 3 jpm-12-01453-f003:**
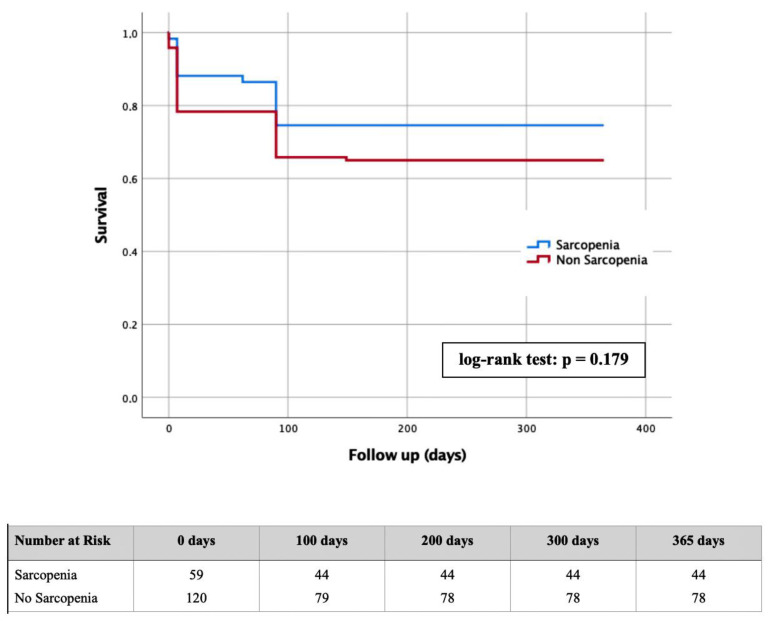
Kaplan-Meier curves with the corresponding number for the detection of one-year survival based on the presence or absence of sarcopenia.

**Figure 4 jpm-12-01453-f004:**
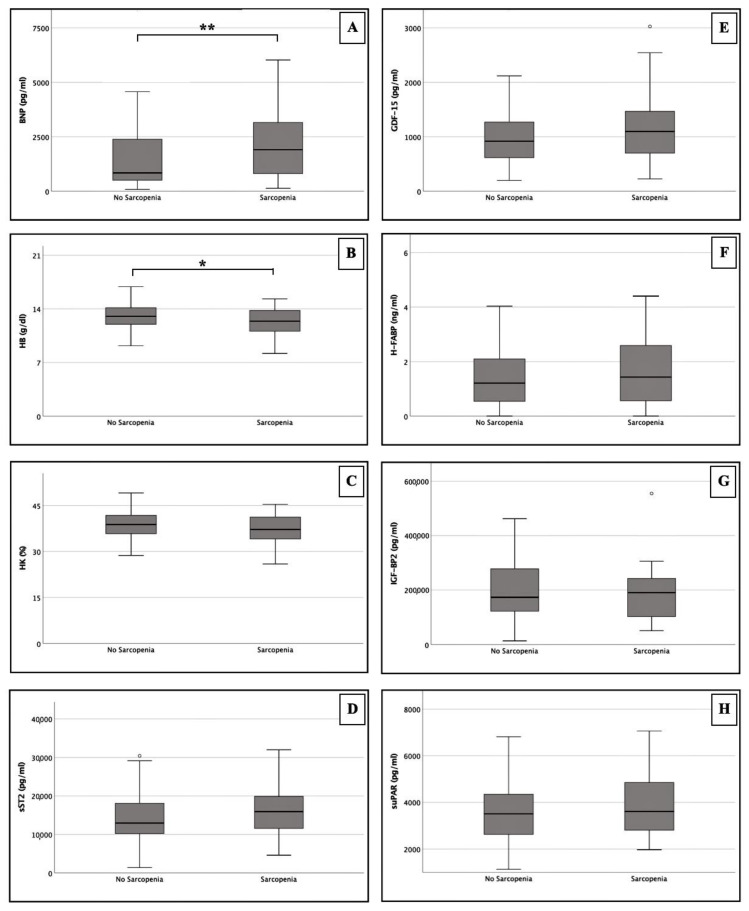
Serum concentrations of BNP (**A**), HB (**B**), HK (**C**), sST2 (**D**), GDF-15 (**E**), H-FABP (**F**), IGF-BP2 (**G**) and suPAR (**H**) in patients with radiological sarcopenia and no radiological sarcopenia; * *p* ≤ 0.05; ** *p* ≤ 0.01; ° outlier. BNP: brain natriuretic peptide; HB: hemoglobin; HK: hematocrit; sST2: soluble suppression of tumorigenicity-2; GDF-15: growth/differentiation of factor-15; H-FABP: heart-type fatty-acid binding protein; IGF-BP2: insulin like growth factor binding protein 2; suPAR: soluble urokinase- type plasminogen activator receptor.

**Table 1 jpm-12-01453-t001:** Baseline characteristics of overall cohort.

	No Sarcopenia *n* = 120	Sarcopenia *n* = 59	
Clinical Data			*p*-value
Age (years)—mean ± SD	82.63 ± 5.05	83.03 ± 4.68	0.610
Gender (male)—%	53.3	52.5	0.921
Weight (kg)—mean ± SD	74.85 ± 15.52	73.72 ± 12.34	0.628
Height (cm)—mean ± SD	165.96 ± 8.74	169.20 ± 9.69	0.026
BMI (kg/m^2^)—mean ± SD	27.18 ± 5.66	25.80 ± 4.19	0.098
NYHA—median ± IQR	3.00 ± 1.00	3.00 ± 1.00	0.352
STSScore—mean ± SD	2.54 ± 1.37	3.19 ± 1.87	0.106
Concomitant Disease			*p*-value
Diabetes mellitus—%	20.0	25.4	0.409
Hypertension—%	81.7	76.3	0.397
CVD—%	79.2	66.1	0.119
CVD—1 vessel—%	23.3	22.0	0.975
CVD—2 vessels—%	10.8	5.1	0.235
CVD—3 vessels—%	10.0	5.1	0.299
Myocardial infarction—%	4.2	1.7	0.397
Atrial fibrillation—%	35.8	30.5	0.480
Pacemaker—%	10.0	3.4	0.122
Malignancy—%	21.7	20.3	0.838
Stroke—%	5.8	3.4	0.468
PAD—%	5.8	5.1	0.838
COPD—%	8.3	10.2	0.686
Echocardiography			*p*-value
LVEF (%)—mean ± SD	55.76 ± 10.40	56.39 ± 11.19	0.713
LVEDD (mm)—mean ± SD	46.90 ± 6.44	44.36 ± 5.76	0.136
IVSd (mm)—mean ± SD	15.38 ± 3.14	15.04 ± 2.69	0.506
AV Vmax (m/s)—mean ± SD	4.33 ± 0.48	4.28 ± 0.68	0.645
AVdPmean (mmHg)—mean ± SD	48.43 ± 12.72	47.79 ± 12.36	0.756
AVdPmax (mmHg)—mean ± SD	76.53 ± 18.74	78.12 ± 19.62	0.606
TAPSE (mm)—mean ± SD	21.86 ± 3.98	21.50 ± 3.47	0.726
sPAP (mmHg)—mean ± SD	44.18 ± 14.59	41.87 ± 17.67	0.421
AVI ≥ II°—%	13.3	23.7	0.085
MVI ≥ II°—%	17.5	15.3	0.705
TVI ≥ II°—%	9.2	20.3	0.044
Laboratory data			*p*-value
Creatinine (mg/dL)—median ± IQR	1.00 ± 0.30	1.00 ± 0.40	0.762
BNP (pg/mL)—median ± IQR	838.00 ± 1887.75	1906.00 ± 2421.08	0.010
cTnI (pg/mL)—median ± IQR	23.50 ± 17.00	23.50 ± 18.00	0.443
HK (%)—median ± IQR	38.80 ± 6.10	37.20 ± 7.70	0.076
HB (g/dL)—median ± IQR	13.05 ± 2.17	12.40 ± 2.90	0.024
CK (U/L)—median ± IQR	83.00 ± 57.00	72.00 ± 92.00	0.313
sST2 (pg/mL)—median ± IQR	12,963.20 ± 8097.40	15,915.15 ± 8448.70	0.117
GDF-15 (pg/mL)—median ± IQR	916.98 ± 682.76	1096.35 ± 789.02	0.158
H-FABP (ng/mL)—median ± IQR	1.21 ± 1.58	1.43 ± 2.05	0.751
IGF-BP2 (pg/mL)—median ± IQR	173,364.04 ± 161,378.51	190,617.55 ± 143,161.16	0.807
suPAR (pg/mL)—median ± IQR	3513.79 ± 1739.57	3609.00 ± 2125.70	0.264

BMI: body mass index; CVD: cardiovascular disease; PAD: peripheral artery disease; COPD: chronic obstructive pulmonary disease; LVEF: left ventricular ejection fraction; LVEDD: left ventricular end-diastolic diameter; IVSd: interventricular septal thickness at diastole; AV Vmax: maximal velocity over aortic valve; AV dpmean: mean pressure gradient over aortic valve; AV dpmax: maximal pressure gradient over aortic valve; TAPSE: tricuspid annular plane systolic excursion; sPAP: systolic pulmonary artery pressure; AVI: aortic valve insufficiency; MVI: mitral valve insufficiency; TVI: tricuspid valve insufficiency; BNP: brain natriuretic peptide; cTnI: cardiac Troponin I; HK: hematocrit; HB: hemoglobin; CK: creatine kinase; sST2: soluble suppression of tumorigenicity-2; GDF-15: growth/differentiation of factor-15; H-FABP: heart-type fatty-acid binding protein; IGF-BP2: insulin like growth factor binding protein 2; suPAR: soluble urokinase- type plasminogen activator receptor.

**Table 2 jpm-12-01453-t002:** Univariate and multivariate binary logistic regression analysis of overall cohort detecting predictors of sarcopenia.

Sarcopenia—Overall	Univariate		Multivariate	
Binary Logistic Regression	Hazard Ratio (95%)	*p*-Value	Hazard Ratio (95%)	*p*-Value
Age	1.017 (0.954–1.085)	0.608		
Height	1.041 (1.004–1.078)	0.028	1.040 (0.996–1.086)	0.072
Weight	0.995 (0.973–1.017)	0.626		
BMI	0.939 (0.871–1.012)	0.097	0.986 (0.907–1.072)	0.743
NYHA	0.745 (0.382–1.452)	0.388		
STSScore	1.297 (0.937–1.794)	0.117		
Diabetes mellitus	1.364 (0.652–2.850)	0.410		
Arterial Hypertension	0.722 (0.338–1.539)	0.398		
Cardiovascular Disease (all)	0.570 (0.280–1.161)	0.122		
CVD—1 vessel	0.988 (0.462–2.113)	0.975		
CVD—2 vessels	0.462 (0.126–1.696)	0.244		
CVD—3 vessels	0.505 (0.136–1.873)	0.307		
Myocardial infarction	0.404 (0.046–3.535)	0.412		
Atrial fibrillation	0.786 (0.403–1.533)	0.480		
Permanent pacemaker	0.316 (0.068–1.460)	0.140		
Malignancy	0.923 (0.428–1.990)	0.838		
Stroke	0.556 (0.112–2.766)	0.474		
PAD	0.865 (0.215–3.472)	0.838		
COPD	1.245 (0.430–3.608)	0.686		
LVEF	1.006 (0.976–1.036)	0.711		
LVEDD	1.080 (0.900–1.297)	0.409		
IVSd	0.962 (0.859–1.077)	0.503		
AV Vmax	0.843 (0.477–1.490)	0.557		
AV dPmean	0.996 (0.971–1.022)	0.754		
AV dPmax	1.004 (0.988–1.022)	0.604		
TAPSE	0.975 (0.849–1.120)	0.721		
AVI ≥ II°	2.026 (0.900–4.564)	0.088	1.975 (0.814–4.793)	0.132
MVI ≥ II°	0.848 (0.362–1.991)	0.706		
TVI ≥ II°	2.443 (1.004–5.942)	0.049	2.029 (0.689–5.978)	0.199
sPAP	0.990 (0.967–1.014)	0.419		
Creatinine	1.090 (0.773–1.537)	0.623		
HB	0.658 (0.469–0.925)	0.016	0.615 (0.416–0.908)	0.015
HK	0.131 (0.020–0.860)	0.034	0.824 (0.299–2.269)	0.709
CK	0.831 (0.566–1.219)	0.343		
BNP	1.415 (1.017–1.969)	0.040	1.131 (0.769–1.663)	0.533
cTnI	2.762 (0.614–12.424)	0.185		
sST2	1.128 (0.816–1.558)	0.466		
GDF-15	1.159 (0.851–1.580)	0.350		
H-FABP	1.007 (0.690–1.469)	0.973		
IGF-BP2	1.147 (0.706–1.865)	0.579		
suPAR	1.196 (0.884–1.617)	0.247		

BMI: body mass index; CVD: cardiovascular disease; PAD: peripheral artery disease; COPD: chronic obstructive pulmonary disease; LVEF: left ventricular ejection fraction; LVEDD: left ventricular end-diastolic diameter; IVSd: interventricular septal thickness at diastole; AV Vmax: maximal velocity over aortic valve; AV dpmean: mean pressure gradient over aortic valve; AV dpmax: maximal pressure gradient over aortic valve; TAPSE: tricuspid annular plane systolic excursion; sPAP: systolic pulmonary artery pressure; AVI: aortic valve insufficiency; MVI: mitral valve insufficiency; TVI: tricuspid valve insufficiency; BNP: brain natriuretic peptide; cTnI: cardiac Troponin I; HK: hematocrit; HB: hemoglobin; CK: creatine kinase; sST2: soluble suppression of tumorigenicity-2; GDF-15: growth/differentiation of factor-15; H-FABP: heart-type fatty-acid binding protein; IGF-BP2: insulin like growth factor binding protein 2; suPAR: soluble urokinase-type plasminogen activator receptor.

**Table 3 jpm-12-01453-t003:** Relevant cardiovascular biomarkers of sarcopenic and non-sarcopenic patients according to gender.

	Male			Female		
Biomarkers Median ± IQR	No Sarcopenia	Sarcopenia	*p*-Value	No Sarcopenia	Sarcopenia	*p*-Value
BNP (pg/mL)	1173.00 ± 2371.53	2167.00 ± 2603.25	0.259	758.00 ± 980.70	1562.50 ± 2450.00	0.007
HB (g/dL)	13.55 ± 2.05	13.50 ± 2.90	0.425	12.70 ± 2.20	11.90 ± 1.88	0.008
HK (%)	40.15 ± 5.75	39.20 ± 7.40	0.631	37.55 ± 5.28	36.15 ± 5.18	0.023
sST2 (pg/mL)	13,520.79 ± 10,191.62	16,677.31 ± 7447.18	0.121	12,521.75 ± 6285.64	14,188.31 ± 7109.88	0.564
GDF-15 (pg/mL)	970.43 ± 582.40	1110.32 ± 924.08	0.446	754.73 ± 803.16	1083.21 ± 685.84	0.158
H-FABP (ng/mL)	1.14 ± 1.79	0.92 ± 1.70	0.754	1.45 ± 1.51	1.66 ± 2.15	0.524
IGF-BP2 (pg/mL)	170,735.87 ± 157,936.82	177,884.49 ± 154,649.85	0.495	178,625.79 ± 185,732.21	216,432.35 ± 149,201.83	0.726
suPAR (pg/mL)	3388.91 ± 1794.63	3556.09 ± 2188.56	0.209	3660.60 ± 1647.93	3970.00 ± 2069.48	0.763

BNP: brain natriuretic peptide; HB: hemoglobin; HK: hematocrit; sST2: soluble suppression of tumorigenicity-2; GDF-15: growth/differentiation of factor-15; H-FABP: heart-type fatty-acid binding protein; IGF-BP2: insulin like growth factor binding protein 2; suPAR: soluble urokinase-type plasminogen activator receptor.

**Table 4 jpm-12-01453-t004:** Univariate and multivariate binary logistic regression analysis of male patients for the detection of predictors of sarcopenia.

Sarcopenia—Male	Univariate		Multivariate	
Binary Logistic Regression	Hazard Ratio (95%)	*p*-Value	Hazard Ratio (95%)	*p*-Value
Age	0.996 (0.918–1.081)	0.925		
Height	1.125 (1.041–1.216)	0.003	1.125 (1.041–1.216)	0.003
Weight	1.005 (0.971–1.039)	0.786		
BMI	0.929 (0.828–1.043)	0.213		
NYHA	0.837 (0.337–2.079)	0.701		
STSScore	0.931 (0.529–1.640)	0.806		
Diabetes mellitus	1.365 (0.498–3.743)	0.546		
Arterial Hypertension	0.874 (0.309–2.470)	0.799		
Cardiovascular Disease (all)	0.597 (0.212–1.678)	0.328		
CVD—1 vessel	0.944 (0.316–2.818)	0.918		
CVD—2 vessels	1.040 (0.240–4.504)	0.958		
CVD—3 vessels	0.725 (0.071–7.347)	0.785		
Myocardial infarction	1.033 (0.090–11.852)	0.979		
Atrial fibrillation	1.204 (0.507–2.860)	0.675		
Permanent pacemaker	0.204 (0.025–1.686)	0.14		
Malignancy	0.745 (0.273–2.032)	0.566		
Stroke	0.644 (0.122–3.392)	0.603		
PAD	0.500 (0.054–4.672)	0.543		
COPD	0.872 (0.210–3.632)	0.851		
LVEF	1.011 (0.972–1.051)	0.591		
LVEDD	0.466 (0.131–1.663)	0.24		
IVSd	0.956 (0.831–1.101)	0.533		
AV Vmax	1.231 (0.500–3.026)	0.651		
AV dPmean	1.010 (0.974–1.048)	0.59		
AV dPmax	1.010 (0.986–1.035)	0.414		
TAPSE	1.016 (0.859–1.202)	0.851		
AVI ≥ II	1.750 (0.496–6.169)	0.384		
MVI ≥ II	0.481 (0.125–1.853)	0.288		
TVI ≥ II	2.280 (0.606–8.584)	0.223		
sPAP	0.991 (0.962–1.022)	0.563		
Creatinine	0.902 (0.525–1.551)	0.709		
HB	0.779 (0.500–1.213)	0.269		
HK	0.318 (0.027–3.723)	0.361		
CK	0.733 (0.441–1.220)	0.232		
BNP	1.169 (0.798–1.711)	0.422		
cTnI	2.190 (0.420–11.426)	0.352		
sST2	1.175 (0.770–1.794)	0.455		
GDF-15	1.170 (0.763–1.793)	0.472		
H-FABP	1.043 (0.502–2.166)	0.91		
IGF-BP2	1.163 (0.695–1.947)	0.566		
suPAR	1.194 (0.842–1.694)	0.319		

BMI: body mass index; CVD: cardiovascular disease; PAD: peripheral artery disease; COPD: chronic obstructive pulmonary disease; LVEF: left ventricular ejection fraction; LVEDD: left ventricular end-diastolic diameter; IVSd: interventricular septal thickness at diastole; AV Vmax: maximal velocity over aortic valve; AV dpmean: mean pressure gradient over aortic valve; AV dpmax: maximal pressure gradient over aortic valve; TAPSE: tricuspid annular plane systolic excursion; sPAP: systolic pulmonary artery pressure; AVI: aortic valve insufficiency; MVI: mitral valve insufficiency; TVI: tricuspid valve insufficiency; BNP: brain natriuretic peptide; cTnI: cardiac Troponin I; HK: hematocrit; HB: hemoglobin; CK: creatine kinase; sST2: soluble suppression of tumorigenicity-2; GDF-15: growth/differentiation of factor-15; H-FABP: heart-type fatty-acid binding protein; IGF-BP2: insulin like growth factor binding protein 2; suPAR: soluble urokinase- type plasminogen activator receptor.

**Table 5 jpm-12-01453-t005:** Univariate and multivariate binary logistic regression analysis of female patients for the detection of predictors of sarcopenia.

Sarcopenia—Female	Univariate		Multivariate	
Binary Logistic Regression	Hazard Ratio (95%)	*p*-Value	Hazard Ratio (95%)	*p*-Value
Age	1.051 (0.946–1.167)	0.356		
Height	1.056 (0.977–1.141)	0.173		
Weight	0.985 (0.951–1.019)	0.38		
BMI	0.945 (0.858–1.041)	0.25		
NYHA	0.597 (0.211–1.688)	0.331		
STSScore	1.962 (1.050–3.668)	0.035	1.923 (0.642–5.756)	0.243
Diabetes mellitus	1.364 (0.463–4.015)	0.574		
Arterial Hypertension	0.574 (0.189–1.750)	0.329		
Cardiovascular Disease (all)	0.533 (0.197–1.443)	0.216		
CVD—1 vessel	1.043 (0.360–3.023)	0.938		
CVD—2 vessels	0.000 (0.000– -)	0.999		
CVD—3 vessels	0.419 (0.084–2.083)	0.288		
Myocardial infarction	0.000 (0.000– -)	0.999		
Atrial fibrillation	0.382 (0.115–1.272)	0.117		
Permanent pacemaker	0.654 (0.065–6.593)	0.719		
Malignancy	1.304 (0.384–4.431)	0.67		
Stroke	0.000 (0.000– -)	1		
PAD	1.359 (0.214–8.640)	0.745		
COPD	2.120 (0.399–11.256)	0.378		
LVEF	0.997 (0.951–1.046)	0.914		
LVEDD	1.111 (0.827–1.492)	0.486		
IVSd	0.975 (0.804–1.182)	0.797		
AV Vmax	0.631 (0.279–1.427)	0.269		
AV dPmean	0.984 (0.950–1.019)	0.362		
AV dPmax	0.999 (0.977–1.023)	0.958		
TAPSE	0.888 (0.689–1.146)	0.362		
AVI ≥ II°	2.167 (0.736–6.378)	0.16		
MVI ≥ II°	1.364 (0.431–4.315)	0.598		
TVI ≥ II°	2.556 (0.765–8.539)	0.127		
sPAP	0.992 (0.954–1.031)	0.685		
Creatinine	1.257 (0.797–1.982)	0.325		
HB	0.471 (0.258–0.858)	0.014	5.782 (0.553–60.504)	0.143
HK	0.024 (0.001–0.630)	0.025	0.386 (0.000–<0.001)	0.95
CK	1.306 (0.540–3.158)	0.553		
BNP	3.274 (1.161–9.232)	0.025	8.737 (0.734–103.958)	0.086
cTnI	10.620 (0.143–791.155)	0.283		
sST2	1.079 (0.639–1.821)	0.776		
GDF-15	1.154 (0.734–1.814)	0.534		
H-FABP	0.985 (0.622–1.559)	0.948		
IGF-BP2	1.042 (0.215–5.044)	0.959		
suPAR	1.201 (0.657–2.194)	0.552		

BMI: body mass index; CVD: cardiovascular disease; PAD: peripheral artery disease; COPD: chronic obstructive pulmonary disease; LVEF: left ventricular ejection fraction; LVEDD: left ventricular end-diastolic diameter; IVSd: interventricular septal thickness at diastole; AV Vmax: maximal velocity over aortic valve; AV dpmean: mean pressure gradient over aortic valve; AV dpmax: maximal pressure gradient over aortic valve; TAPSE: tricuspid annular plane systolic excursion; sPAP: systolic pulmonary artery pressure; AVI: aortic valve insufficiency; MVI: mitral valve insufficiency; TVI: tricuspid valve insufficiency; BNP: brain natriuretic peptide; cTnI: cardiac Troponin I; HK: hematocrit; HB: hemoglobin; CK: creatine kinase; sST2: soluble suppression of tumorigenicity-2; GDF-15: growth/differentiation of factor-15; H-FABP: heart-type fatty-acid binding protein; IGF-BP2: insulin like growth factor binding protein 2; suPAR: soluble urokinase-type plasminogen activator receptor.

## Data Availability

The data presented in this study are available on request from the corresponding author.

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
