# Peer review of "CT-Diagnosed Sarcopenia and Cardiovascular Biomarkers in Patients Undergoing Transcatheter Aortic Valve Replacement: Is It Possible to Predict Muscle Loss Based on Laboratory Tests?—A Multicentric Retrospective Analysis"

_jpm, 2022, doi:10.3390/jpm12091453_

Round 1
Reviewer 1 Report
In the manuscript titled “CT-derived Sarcopenia and Cardiovascular Biomarkers in Patients undergoing TAVR: Searching for the Needle in the Haystack to predict muscle loss based on laboratory tests - a multicentric retrospective analysis”, the authors have demonstrated patients with sarcopenia have no significantly different in cardiovascular biomarker expression compared to patients without muscle loss. This article provides some reference information for clinical work but still lacks novelty. I have the following suggestions to improve the manuscript:
• Due to the small sample size of this study, the results presented still have certain limitations.
• It would be better if the authors could change the table format into a three-line table in an academic paper (Table 1-3). The quality of Table 1, 2 and 3 needs to be improved, the images should be clearer.
• In the part of “Study population”, they said “This study included 179 patients with severe primary degenerative AS planning to undergo TAVR procedure between 2016 and 2018 at University Hospital Salzburg and University Hospital Linz.” The author's statement is not rigorous. It might be more reasonable if the authors could describe the inclusion criteria in more detail and more rigorously.
• I can’t find Figure 1-3, so I hope that the author can add images or point out where they are. The Figure part cannot be reviewed at this time.
• The Limitation section can be integrated into the Discussion section. The author said “No difference in survival between sarcopenic and non-sarcopenic patients”, “In contrast to previous work by Kofler et al. and Saji et al. this result is in clear contradiction.”, the author's statement is not convincing. Since the design of this study is not convincing even to the authors compared to the previous study, why is this study carried out? It may be more convincing if the authors can list more convincing advantages of this study design and the limitations of the previous study compared to this study.
• Please check the typo, punctuation, paragraph structure and grammar throughout the manuscript.
Author Response
Dear Reviewer 1,
thank you very much for the valuable comments. Please see the attached PDF file for a point-by-point response to the comments.
We thank you one more time for a second proofreading and hope that we were able to improve our manuscript.
Kind regards
Elke Boxhammer

Author Response
Dear Reviewer 2,
thank you very much for the valuable comments. Please see the attached PDF file for a point-by-point response to the comments.
We thank you one more time for a second proofreading and hope that we were able to improve our manuscript.
Kind regards
Elke Boxhammer

Round 2
Reviewer 1 Report
After the author's revision, the overall article has been significantly improved.
Reviewer 2 Report
Dear Authors and Editor,
The article has been improved sufficiently.